# Identification of genotype–biochemical phenotype correlations associated with fructose 1,6-bisphosphatase deficiency

Ikki Sakuma[1], Hidekazu Nagano[1], Naoko Hashimoto[1,2], Masanori Fujimoto[1,3], Akitoshi Nakayama[1], Takahiro Fuchigami[1], Yuki Taki[1], Tatsuma Matsuda[1], Hiroyuki Akamine[1], Satomi Kono[1], Takashi Kono[1], Masataka Yokoyama[1], Motoi Nishimura[4], Koutaro Yokote[3], Tatsuki Ogasawara[5], Yoichi Fujii[5], Seishi Ogawa[5,6], Eunyoung Lee[2,7], Takashi Miki[2,7] & Tomoaki Tanaka ◯ [1,2 ✉]

Fructose-1,6-bisphosphatase (FBPase) deficiency, caused by an FBP1 mutation, is an autosomal recessive disorder characterized by hypoglycemic lactic acidosis. Due to the rarity of FBPase deficiency, the mechanism by which the mutations cause enzyme activity loss still remains unclear. Here we identify compound heterozygous missense mutations of FBP1, c.491G>A (p.G164D) and c.581T>C (p.F194S), in an adult patient with hypoglycemic lactic acidosis. The G164D and F194S FBP1 mutants exhibit decreased FBP1 protein expression and a loss of FBPase enzyme activity. The biochemical phenotypes of all previously reported FBP1 missense mutations in addition to G164D and F194S are classified into three functional categories. Type 1 mutations are located at pivotal residues in enzyme activity motifs and have no effects on protein expression. Type 2 mutations structurally cluster around the substrate binding pocket and are associated with decreased protein expression due to protein misfolding. Type 3 mutations are likely nonpathogenic. These findings demonstrate a key role of protein misfolding in mediating the pathogenesis of FBPase deficiency, particularly for Type 2 mutations. This study provides important insights that certain patients with Type 2 mutations may respond to chaperone molecules.

[1] Department of Molecular Diagnosis, Graduate School of Medicine Chiba University, Chiba 260-8670, Japan. [2] Research Institute of Disaster Medicine, Chiba University, Chiba 260-8670, Japan. [3] Department of Endocrinology, Hematology and Gerontology, Graduate School of Medicine Chiba University, Chiba 260-8670, Japan. [4] Division of Laboratory Medicine and Clinical Genetics, Chiba University Hospital, Chiba 260-8670, Japan. [5] Department of Pathology and Tumor Biology, Kyoto University, Kyoto, Japan. [6] Institute for the Advanced Study of Human Biology (WPI-ASHBi), Kyoto University, Kyoto, Japan. [7] Department of Medical Physiology, Chiba University, Graduate School of Medicine, Chiba 260-8670, Japan. ✉email: tomoaki@restaff.chiba-u.jp

Fructose-1,6-bisphosphatase (FBPase) is a key regulatory enzyme in gluconeogenesis that catalyzes the hydrolysis of fructose 1,6-bisphosphate to fructose 6-phosphate and inorganic phosphate[1]. Baker and Winegrad first reported FBPase deficiency in 1970[2], and Kikawa et al. subsequently identified three FBP gene (FBP1) mutations associated with FBPase deficiency in 1997[3]. FBPase deficiency is an autosomal recessive disorder that generally occurs concurrently with hypoglycemia and metabolic acidosis[4]. Although fasting and febrile infections are known to trigger life-threatening episodes of hypoglycemia and lactic acidosis in infancy, these episodes rarely occur in adults due to increased glycogen storage[4]. Several harmful mutations in the coding region of FBP1 have been reported[5]. Nevertheless, with incidence rates estimated to range between 1/350,000 and 1/900,000, FBPase deficiency remains a very rare inherited disease[5,6]. Due to its low incidence, elucidating the molecular mechanism by which the identified mutations cause the loss of enzyme activity is still challenging, particularly in terms of genotype-phenotype correlations and in adult-onset cases. Here, we present the case of a 22-year-old female patient who was diagnosed with FBPase deficiency and exhibited a compound heterozygous mutation of the FBP1 gene. Our results demonstrate that the underlying mechanism by which the identified mutations cause FBPase deficiency involves protein misfolding, and we examined and categorized the biochemical phenotypes of all previously reported FBP1 missense mutations into three functional groups.

## Results

**Case presentation and whole-exome sequencing**. We describe the case of a 22-year-old Japanese female patient who presented with a severe hypoglycemic attack and acidosis induced by prolonged fasting. Blood exams, including a blood gas analysis, showed very low levels of plasma glucose (12 mg/dL), high levels of lactate (110 mg/dL) and severe metabolic acidosis (pH 6.85, $PaCO_2$ 19 mmHg, $HCO_3^-$ 2.5 mmol/L). The patient reported experiencing similar episodes triggered by fasting or febrile illness during her early childhood (Supplementary Table 1). We suspected the case to be an inherited metabolic issue, such as fatty acid oxidation or gluconeogenesis disorders causing hypoglycemic attacks with lactic acidosis, and further examinations were performed to establish a definitive diagnosis. Radiological examination of the abdomen by computerized tomography (CT) revealed a fatty liver, and the liver biopsy showed hepatic steatosis without other alterations (Supplementary Fig. 1a). The acylcarnitine profile exhibited no specific pattern, suggesting a normal β-oxidation pathway. However, the urinary organic acid profile exhibited significantly elevated glycerol, lactate, and pyruvate levels, indicating a possible FBPase deficiency. The oral fructose tolerance test demonstrated a rapid decrease in blood glucose levels and an increase in lactate concentrations (Fig. 1a). In addition, fructose loading increased glycerol levels in the urinary organic acid profile (Fig. 1b). Although clinical findings up to this point strongly indicated a case of FBPase deficiency, a differential diagnosis from other metabolic disorders that occur concomitantly with hypoglycemic attacks and lactic acidosis was difficult due to overlapping clinical features[6].

Previous studies have shown that genetic analyses using next-generation sequencing are useful for the molecular diagnosis of complex metabolic diseases, including FBPase deficiency[7]. Thus, we performed whole-exome sequencing for samples from the proband and her family to identify potential underlying genetic defects. This approach revealed 111 filtered variants in the proband (Supplementary Data 4). Two heterozygous variants were identified in a region of FBP1, whereas there were no variants associated with mitochondrial fatty acid oxidation disorders and gluconeogenesis, such as those in OCTN2, CACT, CPT1, CPT2, LCHAD, MCAD, SCAD, MTP, VLCAD, ACAD9, ETFDH, ETFA, ETFB, HMGCS2, HMGCL, PC, PCK1, PCK2, G6PC, and PGM1. The identified variants in FBP1 were missense mutations (II-1 in Fig. 1c, e); these variants were c.491G > A p.G164D, which is a novel mutation to the best of our knowledge, and c.581T > C p.F194S, which has been previously reported[8]. According to the exome sequencing results of family members, her parents carried a single mutation. The G164D mutation was inherited from the father (I-1 in Fig. 1c, e), and the F194S mutation was inherited from her mother (I-2 in Fig. 1c, e). Then, we confirmed that both FBP1 variants, G164D and F194S, in the proband were heterozygous missense mutations (Fig. 1d) using Sanger sequencing, suggesting that these variants existed in a compound heterozygous state. Based on these findings, we made a definitive diagnosis of FBPase deficiency and advised the patient to avoid both prolonged fasting and the intake of fructose-rich foods under a fasted state.

G164 and F194 are located in the β-strand structure (Fig. 1g). Both of the identified mutations (G164D and F194S) could lead to conformational changes due to the substitution of hydrophobic amino acids with hydrophilic amino acids (Table 1). In general, FBPase catalyzes the hydrolysis of fructose 1,6-bisphosphate to fructose 6-phosphate in the presence of divalent cations, such as magnesium, manganese, or zinc[9], whereas AMP acts as an allosteric inhibitor. The human liver FBPase comprises four identical polypeptide chains. Each of these chains contains 338 amino acid residues, which are assembled as relatively flat tetramers with subunits conventionally labeled C1–C4[9]. These homotetramers consist of two intimate dimers. Figure 1g shows the structure of the FBP1 dimer. We found that neither G164D nor F194S corresponded to pivotal amino acid residues within functional motifs, such as AMP binding sites, metal binding sites, or substrate binding sites (Fig. 1g). Thus, we decided to examine whether and how these mutations affect FBPase enzymatic activity.

**The protein expression and enzymatic activity of the FBP1 mutants were lower than those of the wild-type protein**. We established FBP1 mutants in hepatocytes to analyze molecular function. To eliminate the influence of endogenous FBPase, we generated FBP1-KO HepG2 cells using the CRISPR/Cas9 system (Supplementary Fig. 1b). Next, Myc-tagged FBP1 cDNA constructs of wild-type (WT), mutant clones (G164D or F194S), or cotransfects (G164D and F194S) were transfected into FBP1-KO HepG2 cells. NADP-coupled spectrophotometric assays were used to examine the enzymatic activity of mutant FBPase. As a result, the FBPase activities of all mutants (G164D; 0.73 ± 0.34, F194S; 0.80 ± 0.56, and cotransfected clone; 0.40 ± 0.41 mmol/min/mg protein) were significantly lower than that of the WT clone (4.82 ± 1.61 mmol/min/mg protein) (Fig. 1f). We validated that the transcription level of FBP1 was not greater in the WT clone, but immunoblot analysis revealed that the protein levels of FBP1 mutants were markedly reduced (Fig. 2a, Supplementary Fig. 2). To evaluate the enzymatic activity of the FBP1 protein, we additionally performed an in vitro NADP-coupled spectrophotometric assay with precipitated FBP1 protein by Myc-tag. The results showed that enzyme activity was markedly reduced when the FBP1 protein contained both G164D and F194S (Supplementary Fig. 3a, b).

Subsequently, immunofluorescence analysis was used to evaluate the intracellular localization of the FBP1 gene product. We found that the mutant FBP1 with G164D or F194S

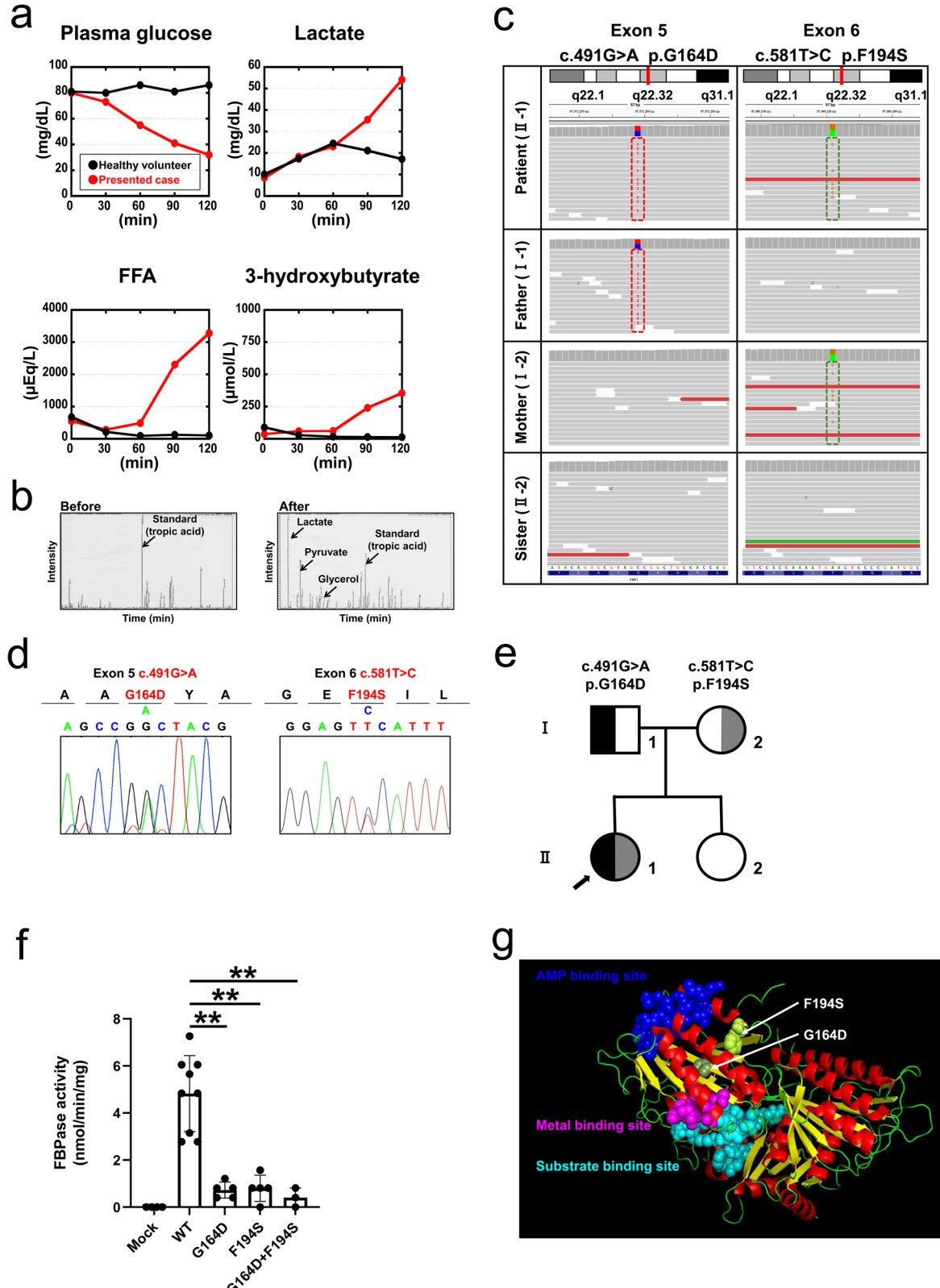

aggregated in the cytoplasm, whereas the FBP1 protein of the WT clone was diffusely localized in the cytoplasm (Fig. 2b). Importantly, this aggregated distribution of FBP1 protein was also found in the liver biopsy from the presented patient (Fig. 2c); therefore, we concluded that the G164D and F194S mutations of *FBP1* cause protein aggregation and the pathogenic loss of enzymatic activity.

**FBP1 proteins with G164D and F194S mutations aggregate in the endoplasmic reticulum due to protein misfolding.** To address the underlying mechanisms of aggregation and reduction in protein expression, we surveyed the interaction partners of FBP1 using liquid chromatography-tandem MS (LC–MS/MS) analysis for the proteins pulled down by immunoprecipitation with an anti-Myc tag. In total, 408 proteins were detected by LC–MS/MS

**Fig. 1 FBP1 mutations identified in an adult patient with severe hypoglycemic acidosis and its clinico-endocrinological profiles.** Panels **a** and **b** demonstrate the results of the oral fructose tolerance test. Oral fructose loading resulted in hypoglycemia accompanied by increased lactate levels. In addition, the 3-hydroxybutyrate levels mildly increased, whereas the free fatty acid (FFA) levels exhibited a robust increase. The patient and a healthy volunteer are shown in red and black, respectively. Panels **c**–**e** show the results of the genetic analyses of the *FBP1* gene. **c** Whole-exome sequencing results of the patient and her family. IGV browser visualization of the whole-exome sequencing results revealed a novel compound heterozygous missense mutation in the *FBP1* gene: c.491G > A p.G164D and c.581T > C p.F194S. The patient inherited the G164D and F194S mutations from her father and mother, respectively. **d** Compound heterozygous missense mutations in the *FBP1* gene, G164D and F194S, were identified in the patient and validated using Sanger sequence analysis. **e** Pedigree and genotypes of the family. Solid symbols indicate the G164D allele, and half-solid symbols indicate the F194S allele. Squares denote males, and circles denote females. The arrow indicates the patient. **f** The FBPase enzyme activity was markedly decreased for the G164D, F194S, and cotransfected constructs compared to that for the WT construct. The data are shown as the mean ± SD. *P < 0.05; **P < 0.01 versus WT (one-way ANOVA test followed by Dunnett's multiple comparison test). Mock: n = 4, WT: n = 9, G164D: n = 5, F164S: n = 5, G164D + F194S: n = 3. **g** Structure of the FBP1 dimer based on the protein data bank (DOI: 10.2210/pdb1FBP/pdb) is shown. Cyan sphere: substrate binding site; magenta sphere: metal binding site; blue sphere: AMP binding site. G164D and F194S are shown as a smudge and in lemon color, respectively.

analysis for the WT, G164D mutant, and F194S mutant (Fig. 2d). Next, we evaluated the molecular functionalities of those proteins interacting with FBP1 and its G164D or F194S mutated variants, including common binding partners among WT and mutants, by Ingenuity Pathway Analysis (IPA) (Fig. 2e). The analysis indicated that FBP1-interacting proteins are mainly involved in protein production and degradation, such as translation, posttranscriptional regulation, the ubiquitination pathway, and the unfolded protein response. To test whether FBP1, especially its mutants, has such a connection to protein production and degradation, we additionally investigated the FBP1 interactome based on the STRING interaction database[10]. Notably, the interactome analysis revealed that the pathways and gene ontology terms associated with unfolded protein binding, heat shock protein binding, protein processing in the endoplasmic reticulum and proteasome were highly enriched (Fig. 2f). Importantly, the FBP1 mutants (G164D and F194S) interacted more strongly with almost all these proteins than WT FBP1 (Supplementary Table 2), suggesting that these molecular pathways are involved in the pathogenesis of FBPase deficiency.

Inherited mutations can disrupt native protein folding, resulting in the formation of misfolded proteins that are consequently retained in the endoplasmic reticulum (ER). These unfolded proteins undergo mannose trimming by ER-associated mannosidase and degradation by the proteasome (endoplasmic reticulum-associated degradation; ERAD), which can be suppressed via the inhibition of ER-associated mannosidase activity using kifunensine[11]. According to previous studies, the protein expression of FBP1 is regulated by the ubiquitin proteasome and autophagy pathways[12]. Thus, to address the involvement of ERAD or protein degradation pathways in FBP1 mutant protein expression, we examined the effects of various inhibitors, such as a proteasome inhibitor (MG-132), an autophagy inhibitor (3-methyladenine), and a potent inhibitor of ER mannosidase I (kifunensine), on the protein expression levels of the G164D and F194S FBP1 mutants. Immunoblot analysis revealed that MG-132 and 3-methyladenine had no impact on the protein expression of these mutants (Supplementary Fig. 3c). In contrast, kifunensine upregulated the protein expression of the G164D and F194S FBP1 mutants, suggesting that these mutants underwent ERAD (Fig. 2i). To further confirm this notion, we performed confocal microscopy analysis to determine the intracellular localization of the G164D and F194S FBP1 mutants, particularly in relation to ER markers. These mutants were primarily, but not completely, found to be colocalized in the ER (Fig. 2g). In addition, immunoblot analyses using the ER fraction revealed that the protein expression of the G164D and F194S FBP1 mutants was markedly higher in the ER than in the whole cell lysate, which is different from what was observed for WT FBP1 (Fig. 2h).

The heat shock proteins (HSPs) are a family of molecular chaperones that collectively form a network that is critical for protein folding[13]. Moreover, missense mutations were shown to shift the folding equilibrium toward a partially folded state, thus increasing the cellular fraction of HSPs relative to the WT protein[14]. Based on these findings, we evaluated the binding of HSP70, HSP90, HSP60, and TCP1 to the FBP1 proteins using anti-Myc immunoprecipitation with subsequent immunoblot analysis (Fig. 2j). The G164D and F194S FBP1 mutants had significant increases in interactions with HSP70, HSP90, HSP60, and TCP1 that were to a greater extent than that in WT FBP1 (Fig. 2k), indicating that these mutants affect the folding equilibrium. Taken together, these findings indicate that the G164D and F194S FBP1 variants caused protein misfolding, which led to a reduction in protein expression and aggregation via the ERAD system.

**Examination of mutation genotype and functional phenotype in all FBP1 missense mutants.** To date, various *FBP1* mutations (14 missense mutations, 12 deletion mutations, four nonsense mutations, four insertions/duplications, two splices, and one indel) have been reported[5]. The reported missense mutations are D119N[7], P120L[5], R158W[6], G164S[3], A177D[3], F194S[8], G207R[5], N213K[15], G260R[16], E281K[17], P284R[8], G294E[18], G294V[15], and V325A[3]. We examined the relationship between the mutated site and the substitution of hydrophobicity in these 14 types of *FBP1* missense mutations and the G164D mutation we newly identified in this patient (Fig. 3a). Eight mutations (G164D, G164S, A177D, F194S, G207R, G260R, P284R, and G294E) exhibited substitutions of hydrophobic amino acids with hydrophilic amino acids, and one mutation (R158W) exhibited a substitution of a hydrophilic amino acid with a hydrophobic amino acid. These nine mutations, defined as mutations that change hydrophobicity, were not located at key amino acid residues (Fig. 3a, indicated by arrowheads) within functional motifs of the substrate binding site, metal binding site, or AMP binding site. Six mutations (D119N, P120L, N213K, E281K, G294V, and V325A) did not generate any change in amino acid hydrophobicity, and four of these mutations (D119N, P120L, N213K, and E281K) were directly located at important amino acid residues of enzyme activity [D119N and E281K were located at key residues in the metal binding site; P120L was located in the linker lesion between the metal binding site (D119 and L121) and the substrate binding site (D122); and N213K was located in the substrate binding region (N213-Y216)] (Fig. 3a, indicated by arrowheads).

Based on this information, we assessed how the alteration of hydrophobicity affects enzyme activity and protein aggregation. We generated Myc-tagged *FBP1* cDNA constructs of mutant clones (D119N, P120L, R158W G164S, A177D, G207R, N213K, G260R, E281K, P284R, G294E, G294V, and V325A) in addition to patient's mutant clones (G164D or F194S). These *FBP1* cDNA constructs were transfected into *FBP1*-KO HepG2

**Table 1 Categories of FBP1 missense mutations.**

| Mutation | Location | Enzymatic activity | Protein expression | Intracellular Localization | Aggregation | Chaperone binding ability | | Type of amino acid | | Hydropathy index | | Structure |
|---|---|---|---|---|---|---|---|---|---|---|---|---|
| Amino acid change | | Compared to wild-type | Compared to wild-type | Localization | (%) | HSP70 | HSP90 | Wild-type | Mutant | Wild-type | Mutant | |
| Type 1: Loss of enzyme activity due to mutations in functional domain with unchanged protein expression and diffuse cytoplasmic localization | | | | | | | | | | | | |
| D119N | Metal binding site | Decrease | No change 1.4 | Diffuse (cytoplasm) | 9.4 | 0.1 | 0.3 | Hydrophilic-acidic | Hydrophilic-neutral | −3.5 | −3.5 | β-strand |
| P120L | Linker lesion at metal and substrate binding site | Decrease | No change 1.3 | Diffuse (cytoplasm) | 8.5 | 1.1 | 0.5 | Hydrophobic-aliphatic | Hydrophobic-aliphatic | −1.6 | 3.8 | β-strand |
| N213K | Substrate binding site | Decrease | No change 1.1 | Diffuse (cytoplasm) | 13.3 | 1.5 | 0.5 | Hydrophilic-neutral | Hydrophilic-basic | −3.5 | −3.9 | ND |
| E281K | Metal binding site | Decrease | No change 0.9 | Diffuse (cytoplasm) | 33.6 | 2.4 | 0.5 | Hydrophilic-acidic | Hydrophilic-basic | −3.5 | −3.9 | turn |
| Type 2: Loss of enzyme activity with decreased protein expression and ER aggregation in association with hydrophobicity change | | | | | | | | | | | | |
| R158W | | Decrease | Decrease 0.5 | Aggregated (in ER) | 69.6 | 7.7 | 7.8 | Hydrophilic-basic | Hydrophobic-aromatic | −4.5 | −0.9 | α-helix |
| G164D | | Decrease | Decrease 0.4 | Aggregated (in ER) | 71.4 | 13.7 | 15.8 | Hydrophobic-aliphatic | Hydrophilic-acidic | −0.4 | −3.5 | β-strand |
| G164S | | Decrease | Decrease 0.6 | Aggregated (in ER) | 42.7 | 6.4 | 12.2 | Hydrophobic-aliphatic | Hydrophilic-neutral | −0.4 | −0.8 | β-strand |
| A177D | | Decrease | Decrease 0.2 | Aggregated (in ER) | 73.9 | 24.0 | 37.5 | Hydrophobic-aliphatic | Hydrophilic-acidic | 1.8 | −3.5 | β-strand |
| F194S | | Decrease | Decrease 0.4 | Aggregated (in ER) | 62.6 | 15.4 | 19.8 | Hydrophobic-aromatic | Hydrophilic-neutral | 2.8 | −0.8 | β-strand |
| G260R | | Decrease | Decrease 0.4 | Aggregated (in ER) | 73.5 | 18.9 | 28.2 | Hydrophobic-aliphatic | Hydrophilic-basic | −0.4 | −4.5 | ND |
| P284R | | Decrease | Decrease 0.3 | Aggregated (in ER) | 57.7 | 41.2 | 47.2 | Hydrophobic-aliphatic | Hydrophilic-basic | −1.6 | −4.5 | α-helix |
| G294E | | Decrease | Decrease 0.4 | Aggregated (in ER) | 64.1 | 64.9 | 89.9 | Hydrophobic-aliphatic | Hydrophilic-acidic | −0.4 | −3.5 | ND |
| G294V | | Decrease | Decrease 0.6 | Aggregated (in ER) | 65.9 | 6.1 | 3.5 | Hydrophobic-aliphatic | Hydrophobic-aliphatic | −0.4 | 4.2 | ND |
| Type 3: Likely non-pathogenic mutations in regard to biochemical phenotype | | | | | | | | | | | | |
| G207R | | No change | No change 1.0 | Diffuse (cytoplasm) | 12.7 | 3.1 | 3.9 | Hydrophobic-aliphatic | Hydrophilic-basic | −0.4 | −4.5 | ND |
| V325A | | No change | No change 1.0 | Diffuse (cytoplasm) | 25.4 | 2.3 | 1.3 | Hydrophobic-aliphatic | Hydrophobic-aliphatic | 4.2 | 1.8 | α-helix |

*ND* no data, *ER* endoplasmic reticulum.

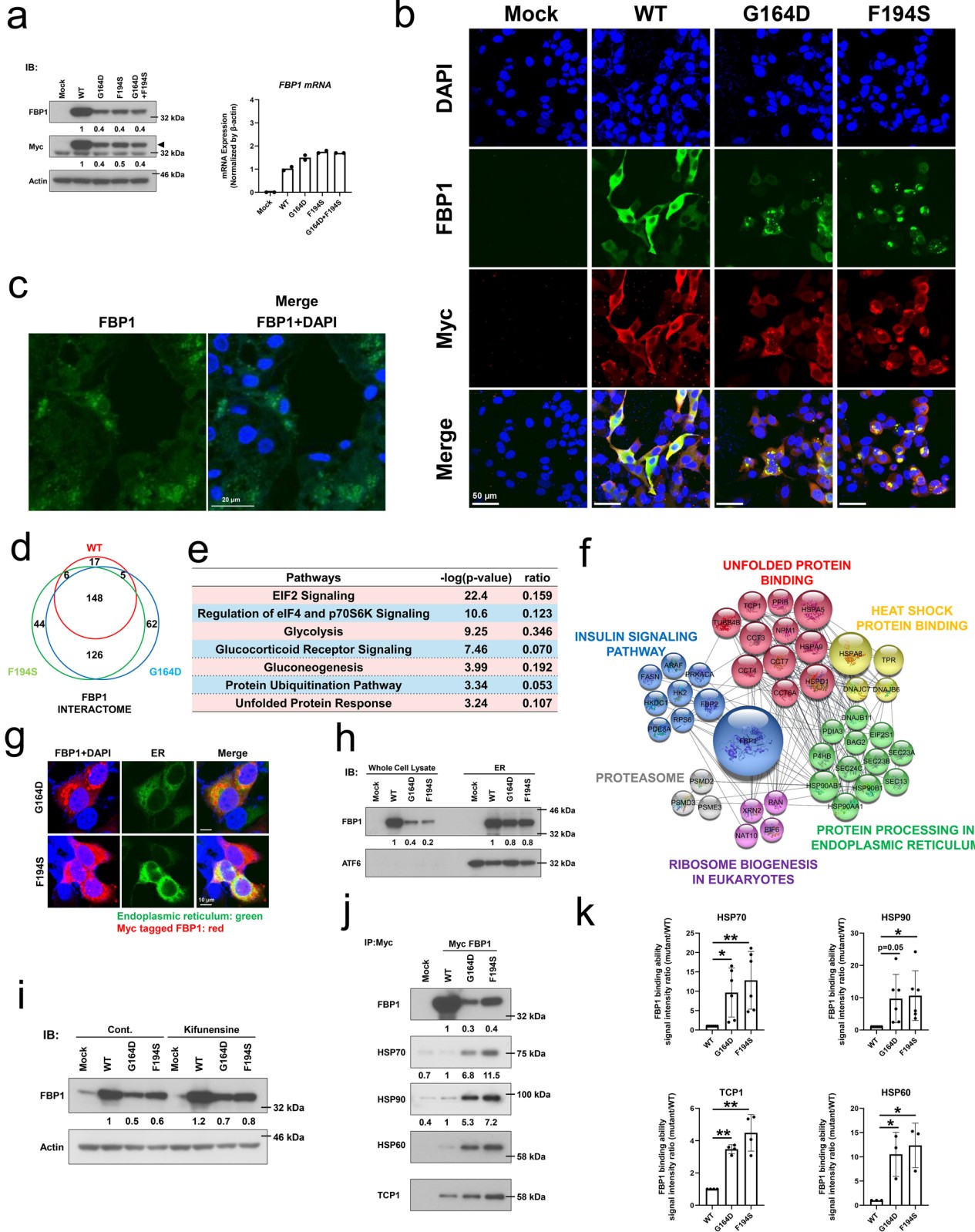

cells with subsequent in vitro functional analysis. Then, these clones were separated into two groups: six mutants without a change in hydrophobicity (indicated in blue) and nine mutants with a change in hydrophobicity (indicated in red) (Fig. 3). We first validated the similar gene transfection in each clone by qPCR (Supplementary Fig. 4). Although two mutants, G207R and V325A, maintained enzyme activity compared to the WT

clone, most clones demonstrated a loss of FBPase activity (Fig. 3b).

At the protein level, there were several exceptions, but FBP1 expression tended to be different depending on the substitution of hydrophobicity. The mutations without changes in hydrophobicity (indicated in blue), except G294V, maintained a similar level of protein expression as WT FBP1, whereas the mutations with

**Fig. 2 Biochemical consequences of the G164D and F194S mutations of FBP1 in association with decreased protein expression and aggregation in ER due to protein misfolding.** In vitro functional analysis of the G164D and F194S FBP1 mutants in *FBP1*-KO HepG2 cells. FBP1 protein expression was examined. **a** Protein expression analysis demonstrated that Myc-tagged FBP1 was decreased in G164D, F194S, and cotransfected constructs compared with that in the WT construct. RT–qPCR analysis detected sufficient quantities of mRNA among the constructs ($n = 2$). **b** Immunofluorescence analysis showed that G164D and F194S FBP1 mutants resulted in the aggregation of FBP1 in the cytoplasm. The scale bars indicate 50 μm. **c** Immunofluorescence analysis of a liver biopsy specimen from the patient. FBP1 aggregated in the cytoplasm. The scale bar indicates 20 μm. **d** Venn diagram of the proteins identified in the FBP1 interactome based on MS analysis. **e** The main function of FBP1 interactome proteins based on Ingenuity Pathway Analysis (IPA). **f** Clusters of the enriched biological processes in the FBP1 interactome. Data from the STRING interaction database were visualized using Cytoscape. **g** Intracellular localization of the endoplasmic reticulum (green) and Myc-tagged FBP1 (red). The G164D and F194S FBP1 mutants colocalized, but not completely, in the endoplasmic reticulum. The scale bars indicate 10 μm. **h** Immunoblot analyses of FBP1 and ATF6. ATF6 is an endoplasmic reticulum, stress-regulated, transmembrane transcription factor. It was enriched in the endoplasmic reticulum fractions. FBP1 protein was detected in the whole cell lysate and endoplasmic reticulum fractions. Protein expression of the G164D and F194S FBP1 mutants was relatively high in the endoplasmic reticulum compared to the whole cell lysate. **i** HepG2 cells were treated with the mannosidase inhibitor kifunensine (200 μM) for 48 h, which increased the protein expression of the G164D and F194S FBP1 mutants. **j** Immunoprecipitation with anti-Myc and immunoblot analysis of FBP1, HSP70, HSP90, HSP60, and TCP1. **k** G164D and F194S FBP1 mutants demonstrated greater interactions with HSP70, HSP90, HSP60 and TCP1 compared with wild-type (WT) FBP1. The data are presented as the mean ± SD. *$P < 0.05$; **$P < 0.01$ versus WT (one-way ANOVA test followed by Dunnett's multiple comparison test). WT: $n = 3$–6, G164D: $n = 3$–6, F164S: $n = 3$–6. Vertically stacked strips of bands in in (**a**, **h**, **i**, and **j**) were evaluated in the same experimental conditions respectively while they were not in fact all derived from the same gel.

changes in hydrophobicity (indicated in red), except G207R, exhibited decreased protein expression (Fig. 3c, d). Consistent with the protein expression, immunofluorescence staining demonstrated that all mutants with protein expression similar to WT were diffusely localized in the cytoplasm, whereas mutants with decreased protein expression aggregated in the cytoplasm (Fig. 4a, b). Subsequently, we observed a strong negative correlation between the number of cells with FBP1 aggregates and protein expression among the FBP1 missense mutants (Fig. 4c). These data suggested that protein aggregation could be linked to the substitution of hydrophobicity, and FBPase activities were more broadly affected by missense mutations independent of hydrophobicity status. As there were several exceptions, including G207R and V325A, the other constructive feature could be associated with enzymatic function.

**FBP1 missense mutations were categorized into three functional phenotypes.** Based on these results, we categorized the *FBP1* missense mutations into three functional groups (Table 1, Fig. 5). The Type 1 mutations (D119N, P120L, N213K, and E281K) are direct substitutions of pivotal amino acid residues inside the enzyme activity site (Fig. 5a, b). These mutations did not change their amino acid hydrophobicity or protein expression and were diffusely localized in the cytoplasm, similar to WT FBP1 (Table 1). Therefore, these mutations cause a primary loss of FBPase enzymatic activity through mutations of key amino acid residues in the functional motif of enzymatic activity (indicated by arrowheads) without affecting protein expression and cytoplasmic localization. Indeed, Type 1 mutations are characterized by no change in hydrophobicity.

Type 2 mutations (R158W G164D, G164S, A177D, F194S, G260R, P284R, G294E, and G294V) are likely to be located outside of the important amino acid residues in the functional motif (Fig. 5a) and appear to cluster around the substrate binding pocket (Fig. 5c). In line with the changes in amino acid hydrophobicity (except for G294V), Type 2 mutations decreased protein expression and caused aggregation in the cytoplasm, possibly due to protein misfolding.

Regarding G294V, the hydropathy index shows certain amount of change between glycine (−0.4) and valine (4.2), even though both glycine and valine are hydrophobic amino acids (Table 1).

Type 3 mutations (G207R and V325A) were structurally distant from sites associated with the enzyme activity motif and substrate binding pocket (Fig. 5d). These mutations exhibited normal FBPase enzyme activity, and their protein expression and

cytoplasmic localization were similar to those of WT FBP1 regardless of amino acid hydrophobicity. These findings indicate that Type 3 mutations are likely non-pathogenic in terms of the biochemical phenotype of FBPase deficiency. In fact, previous reports suggested that cases with V325A mutations had no functional defect[3].

As expected, when we examined the binding ability of all mutants to HSP, the interaction with HSP70 and HSP90 was increased to a greater extent for only Type 2 mutants compared with that seen for WT FBP1 and Type 1 and Type 3 mutants (Fig. 5e and Supplementary Fig. 5). Furthermore, the binding ability of WT FBP1 and the FBP1 mutants to either HSP70 or HSP90 was significantly correlated with the number of cells with FBP1 aggregates (%) (Fig. 5f). Thus, we postulate that decreased protein expression due to protein misfolding in association with HSP recognition and protein aggregation via the ERAD system is involved in the pathogenesis of FBPase deficiency, particularly for Type 2 mutations.

## Discussion
FBPase deficiency is sometimes misdiagnosed or diagnosed in delay following initial suspicion of other energy-related deficiencies. Early diagnosis is very important in FBPase deficiency[19]. Our patient's case and the observations in the present study illustrate how FBPase deficiency can be confused with disorders of fatty acid oxidation. While acylcarnitine profiles are useful for the diagnosis of the latter, we advocate that fructose tolerance tests and genetic analyses would be beneficial and provide informative confirmation for the diagnosis of patients with FBPase deficiency.

In general, fructose loads can lead to large, rapid expansions in the hexose- and triose-phosphate pools, potentially providing increased substrates for all central carbon metabolic pathways, including glycolysis, glycogenesis, gluconeogenesis, lipogenesis, and oxidative phosphorylation. In FBPase deficiency, a marked decrease in intracellular free phosphate due to hepatic accumulation of fructose 1,6-bisphosphate can inhibit glycogenolysis, leading to fructose-induced hypoglycemia.

To date, several harmful mutations in the coding region of FBP1 have been reported. Santer et al. reviewed 35 different mutations, including 14 missense mutations, 12 deletion mutations, four nonsense mutations, four insertions/duplications, two splices, and one indel[5]. Although the FBP1 c.581 T > C (which results in the missense mutation F194S) and c.490 G > A (which affects the neighboring nucleotide) mutations have been previously

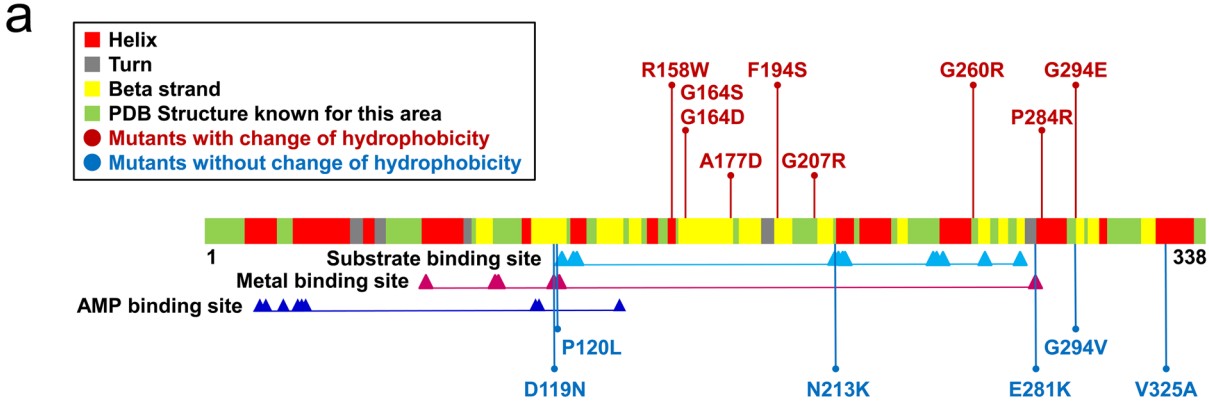

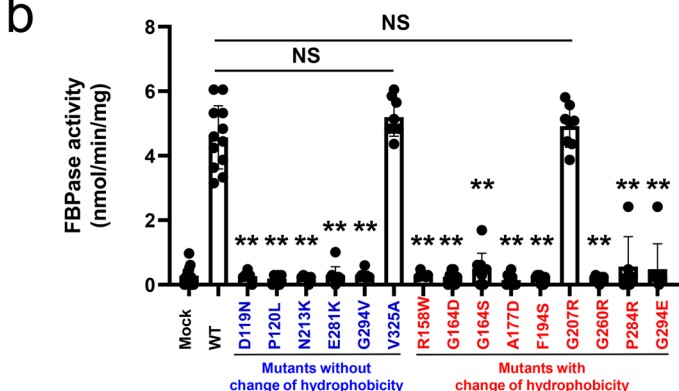

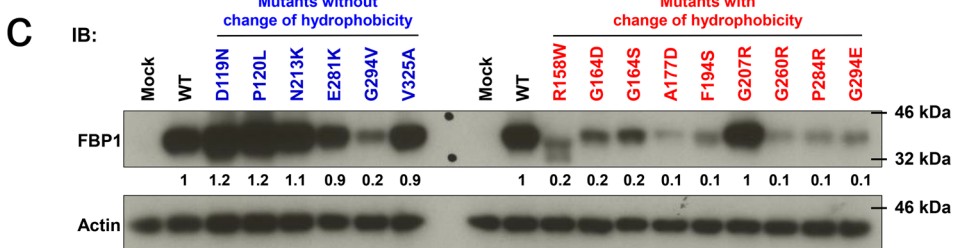

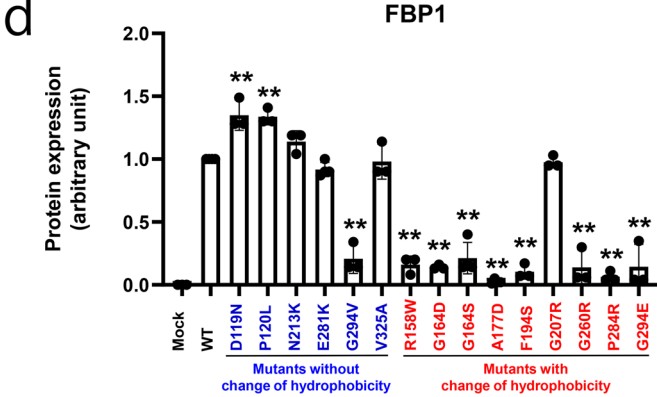

demonstrated[3,8], the c.491 G > A mutation (responsible for the missense mutation G164D) identified in this study is novel to the best of our knowledge. The G164D and F194S mutations, which are located in the β-strand (Figs. 1g and 3a), demonstrated the substitution of hydrophobic amino acids with hydrophilic amino acids (Table 1), suggesting certain conformational changes. The examinations carried out in this study confirmed that the G164D

and F194S mutations of FBP1 resulted in decreased protein expression and a loss of FBPase enzyme activity.

The interactome analysis based on liquid chromatography-tandem MS data for binding partners demonstrated that FBP1, particularly in its mutant forms, interacts with molecular chaperone proteins related to the unfolded protein response, including HSPs. Protein misfolding has been recognized as an

**Fig. 3 Characterization of the enzyme activity and protein expression of all previously reported FBP1 missense mutations. a** Schematic distribution of all previously reported *FBP1* missense mutations. Substrate binding site, metal binding site, and AMP binding sites are shown according to the NCBI and UniProt databases. **b** FBPase enzyme activity of all FBP1 missense mutants. All these mutants, except for G207R and V325A (NS: not significant), exhibited a loss in enzymatic activity (**$P < 0.01$ versus WT) (one-way ANOVA test followed by Dunnett's multiple comparison test). The data are presented as the mean ± SD; $n = 6$–12. **c, d** Protein expression of all FBP1 missense mutants in *FBP1*-KO HepG2 cells. All the mutants that did not change their hydrophobicity, except for G294V, exhibited sufficient protein expression compared with that of WT FBP1, whereas all the mutants that changed their hydrophobicity, except for G207R, exhibited decreased protein expression. The data are presented as the mean ± SD. **$P < 0.01$ versus WT (one-way ANOVA test followed by Dunnett's multiple comparison test) ($n = 3$–4). Vertically stacked strips of bands in a figure were evaluated in the same experimental conditions respectively while they were not in fact all derived from the same gel.

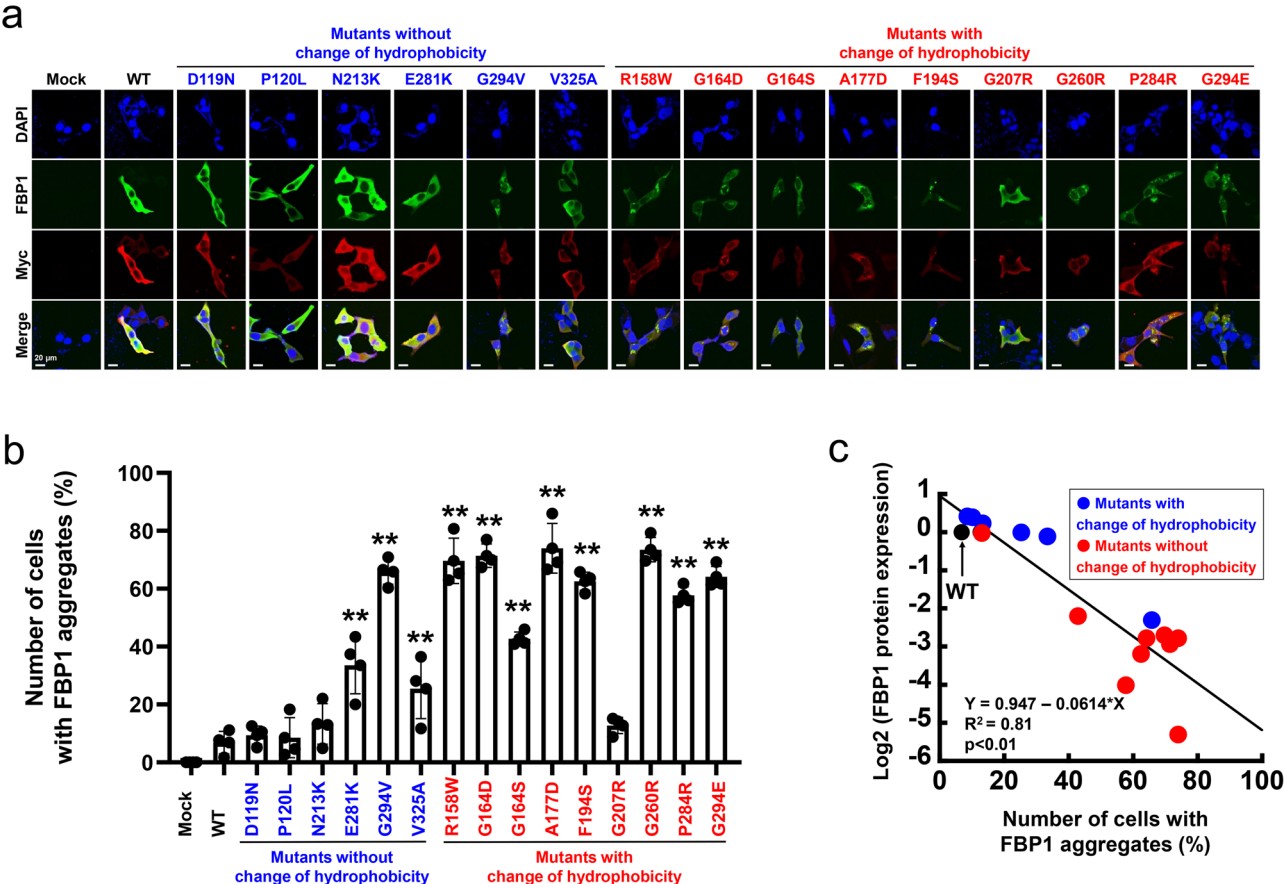

**Fig. 4 Intracellular localization of FBP1 missense mutations in FBP1-KO HepG2 cells. a, b** Immunofluorescence studies of FBP1 missense mutants are shown. The scale bars indicate 20 μm. We counted the number of cells in two groups: diffuse localization of intracellular FBP1 and cytoplasmic aggregation of intracellular FBP1. Then, the ratio of cells with FBP1 aggregates (%) was calculated. All mutants in which hydrophobicity was unchanged, except for G294V, were diffusely localized in the cytoplasm similar, which is similar to the localization of WT FBP1. However, all mutations in which hydrophobicity was changed, except for G207R, were aggregated in the cytoplasm. The data are presented as the mean ± SD. *$P < 0.05$; **$P < 0.01$ versus WT (one-way ANOVA test followed by Dunnett's multiple comparison test) ($n = 4$). **c** A negative correlation was observed between the number of cells with FBP1 aggregates (%) and FBP1 protein expression (mutant/WT). The correlation was determined by the Pearson coefficient of determination, $R^2$.

important pathophysiological cause of protein deficiency in some genetic disorders, such as Fabry disease, Pompe disease, and Gaucher disease[20]. Inherited mutations can disrupt native protein folding, resulting in the formation of misfolded proteins that are consequently retained in the endoplasmic reticulum (ER). HSPs are a family of molecular chaperones that collectively form a network that is critical for protein folding[13]. It has been suggested that HSP70 recognizes unfolded proteins, and HSP90 recognizes partially folded proteins. However, fully folded proteins do not bind with either HSP70 or HSP90[14]. In addition, missense mutations have been shown to shift the protein folding equilibrium toward a partially folded state, thus increasing the cellular fraction of HSP70 and HSP90 relative to WT proteins[14]. Some proteins can only be partially folded by HSP70 and therefore

require additional assistance from HSP60 (chaperonin) to acquire a folded functional conformation[13]. Aberrant protein aggregation was found to be controlled by chaperonins containing TCP-1. We found that the G164D and F194S FBP1 variants increase the interactions between FBP1 and HSP70, HSP90, HSP60, and TCP1. In addition, the FBP1 proteins are partially aggregated and trapped in the ER. Thus, we conclude that the decrease in protein expression detected in FBPase deficiency is, at least in part, a result of protein misfolding.

Finally, 15 FBP1 missense mutations (Figs. 3, 4, and 5) were reviewed and classified into three categories. Type 1 mutations cause a loss of enzyme activity due to mutations in the functional domain. For these mutations, protein expression is unchanged and presents a diffuse cytoplasmic localization. Type 1 mutations

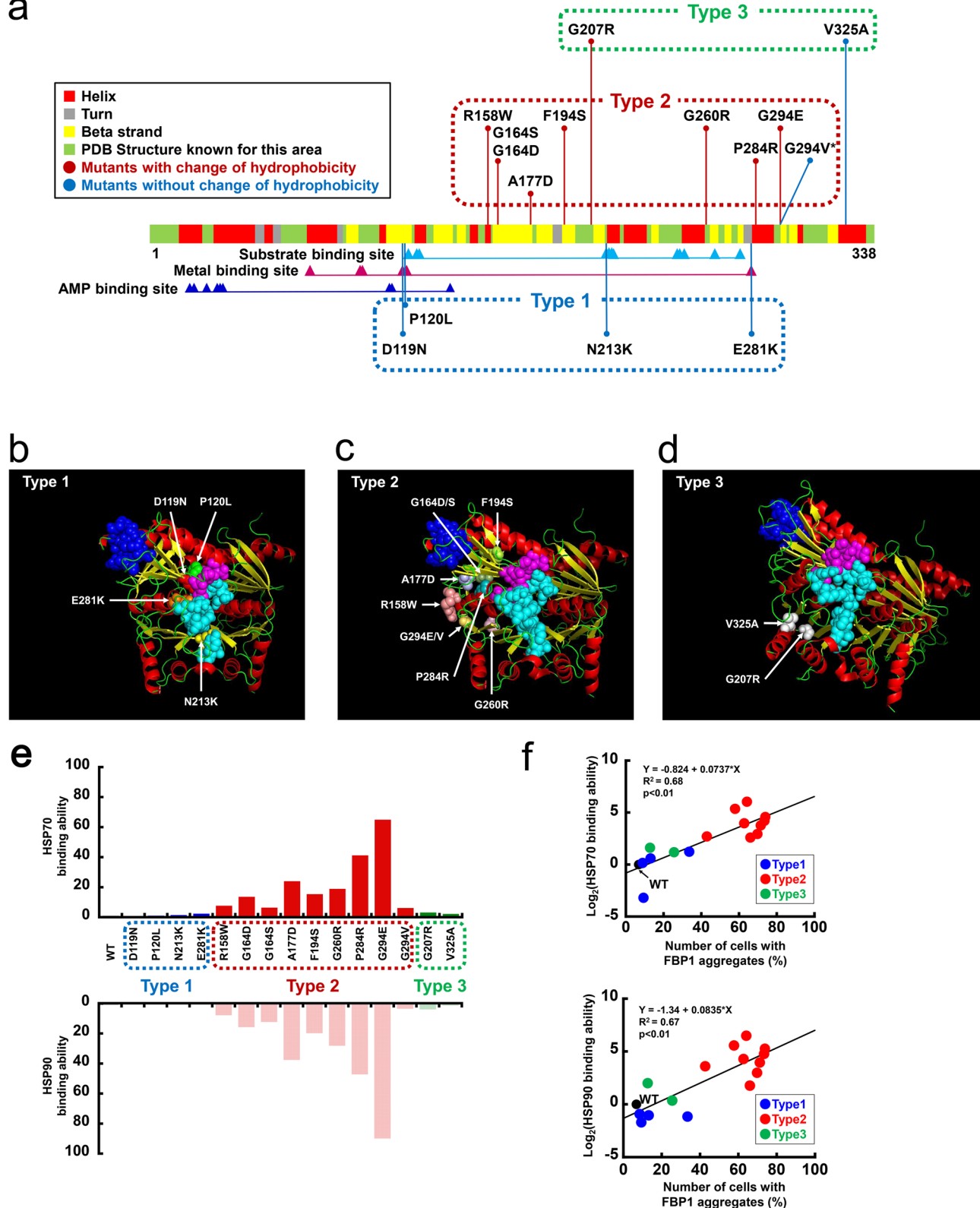

do not change their amino acid hydrophobicity. Type 2 mutations cause a loss of enzyme activity with a reduction in protein expression and ER aggregation due to protein misfolding. Type 2 mutations change amino acid hydrophobicity, except for G294V. Both glycine and valine are classified as hydrophobic amino acids. Therefore, the G294V variant was grouped as a variant without a change in hydrophobicity in Fig. 3. However, there is a certain

change in the hydropathy index between glycine (−0.4) and valine (4.2) (Table 1). This hydropathy difference among hydrophobic amino acids may affect FBP1 protein misfolding. Type 3 mutations are likely non-pathogenic mutations in regard to the obvious phenotype of FBPase deficiency, and the relationship between these mutations and disease onset is obscure. The G207R mutation, wherein the second allele exhibited a

**Fig. 5 Categorization of FBP1 missense mutations based on genotype-functional phenotype associations. a** Schematic distribution of all previously reported *FBP1* missense mutations. Type 1 mutations, which are those without a change in amino acid hydrophobicity, are direct substitutions of key residues within an enzymatically active site. Type 2 mutations are those with a change in hydrophobicity, except for G294V, and Type 3 mutations are those that are not located at pivotal residues in functional motifs. **b**–**d** Structure of the FBP1 dimer based on the protein data bank (DOI: 10.2210/pdb1FBP/pdb). Locations of the mutations are shown. **b** Type 1 mutations include D119N (red), P120L (green), N213K (yellow), and E281K (orange). These mutations are directly located at pivotal residues in functional sites of substrate or metal binding sites. **c** Type 2 mutations include R158W (salmon), G164D/S (smudge), A177D (light blue), F194S (lemon), G260R (light pink), P284R (teal), and G294E/V (yellow orange). These mutations are likely to be located outside of the important amino acid residues in the functional motif and appear to cluster around the substrate binding pocket. **d** Type 3 mutations include G207R (white) and V325A (white), and these mutations are structurally distant from the sites associated with the enzyme activity motif and substrate binding pocket. **e** Binding ability of HSP70 and HSP90 to FBP1 mutant proteins. Type 2 mutants showed increased HSP70 and HSP90 interactions compared to WT FBP1 and Type 1 and Type 3 mutants. **f** Scatterplot of the signal intensity ratios of HSP70 (upper) or HSP90 (lower) binding ability (mutant/WT) and the number of cells with FBP1 aggregates (%). Both HSP70 and HSP90 binding ability were significantly correlated with FBP1 aggregates.

deletion of exon 8, was present in FBPase deficiency[5]. This variant has been previously reported in the heterozygous state with an allele frequency of 0.0001498 in 10 European (non-Finnish) individuals[5], although its pathological effects have not yet been elucidated. In contrast, the V325A mutation is not recorded in the dbSNP database. Although this mutation was observed in patients with FBPase deficiency, it was found to harbor the same mutant alleles as G164S. Based on an examination of the chimeric V325A mutation, Kikawa et al. suggested that this mutation does not play a pathogenic role in FBPase deficiency[3], which is consistent with the findings of our study. Thus, G207R and V325A were defined as Type 3 mutations.

This finding is consistent with clinical characteristics, but further investigation is needed to uncover the role of our defined sites.

In the context of genotype–biochemical phenotype associations, previous in vitro studies evaluated only two missense mutations (D119N and G164S), and consistent with our findings, D119N decreased enzymatic activity with no impact on protein expression[7]. However, G164S decreased protein expression[21]. In addition to missense mutations, several types of *FBP1* mutations (12 deletion mutations, four nonsense mutations, four insertions/duplications, two splices, and one indel) have been reported[5]. Previous in vitro studies reported that c.704delC, c.838delT, and c.960dupG all decreased truncated FBP1 protein expression[7,21,22]. However, the mechanism that leads to a decrease in truncated FBP1 protein expression has not been sufficiently elucidated. Considering our findings, protein misfolding may be involved in the underlying disease pathophysiology in these truncated types of *FBP1* mutations, similar to Type 2 mutations.

The clinical applications of pharmacological chaperone therapy for Fabry disease have become increasingly popular. The chaperone molecules support the folding of mutated enzymes and increase their stability and activity, which may be broadly applicable to other protein deficiencies. Life-threatening episodes of hypoglycemia and lactic acidosis are sometimes triggered by fasting and febrile infections during childhood in cases of FBPase deficiency. However, to date, no preventive agent has been developed for these patients and their symptoms. Our findings indicate the possibility that certain patients with Type 2 mutations may respond to such chaperone molecules, although Type 1 mutations are likely to be untreatable cases of FBPase deficiency. Further studies are warranted to fully understand the pathophysiology of this rare disease as well as to clarify the usefulness and efficacy of pharmacological chaperone therapy.

## Methods

**Mutation analysis**. DNeasy Blood and Tissue Kits (QIAGEN, Hilden, Germany) were used to extract genomic DNA from the patient's blood. Then, all 7 exons of the *FBP1* gene were amplified by polymerase chain reaction (PCR) and sequenced using a 3130 Genetic Analyzer (Applied Biosystems, Massachusetts, USA). Supplementary Table 3 describes the primer information and PCR conditions.

**Whole-exome sequencing**. Whole-exome sequencing involved the targeted capture of all exon sequences using SureSelect Human All Exon v6 (Agilent Technologies, California, USA), followed by massive parallel sequencing of the enriched exon fragments on the HiSeq 2500 platform (Illumina, California, USA) using the 125-bp paired-end mode as per the manufacturer's protocol[23]. The sequenced reads were aligned to a human genome reference (GRCh37) using the default parameter settings in Burrows−Wheeler Aligner version 0.7.10, and the PCR duplicates were eliminated using Picard-tools version 1.39 (http://picard.sourceforge.net/). Candidate mutations with (i) depths ≥8, (ii) number of variant reads ≥4, and (iii) variant allele frequencies (VAFs) of 0.4–0.6 and 1 were adopted, as VAFs of pathogenic germline mutations are estimated to be approximately 0.5 in the heterozygous state and 1 in the homozygous state. Mutations were further filtered by excluding (i) variants presenting only in unidirectional reads, (ii) insertions and deletions in simple repeat regions, (iii) synonymous SNVs, and (iv) known variants listed in the 1000 Genomes Project (Nov 2010 release), Exome Sequencing Project (ESP) 6500, Human Genome Variation Database (HGVD; October 2013 release), and ExAC database with frequencies >0.001 to exclude non-pathogenic variants.

The quality control process for the whole-exome sequencing data is shown in the Supplementary Data 1. For the whole-exome sequencing of the patient and her family, the average depths were 132–150. At least 10x coverage for affected individuals was achieved with 127–143 million reads of 241–245 bp, which is sufficient for consistent breadth of coverage across the exome according to a previous article[24].

**Construction of FBP1 expression vectors**. Total RNA was extracted from the patient's blood using RNeasy Kits (QIAGEN, Hilden, Germany) and reverse-transcribed using SuperScript 2 reverse transcriptase (Thermo Scientific, Massachusetts, USA) and oligo (dT) primers. Amplification of the coding region of the *FBP1* gene was carried out using the forward primer 5′-CACCATGGCTGACCAGGCGCCCTTCG-3′ and the reverse primer 5′-TCACTGGGCAGAGTGCTTCTCATAC-3′. The PCR procedure consisted of the following steps: (a) denaturing at 98 °C for 2 min, followed by 94 °C for 15 secs; and (b) annealing for 30 secs at 58 °C and extension at 72 °C for 1 min for 30 cycles. The PCR products were subcloned into the pGEM-T Easy Vector (Promega, Wisconsin, USA). Then, the *FBP1* fragments (WT and 2 mutants) were cut from the pGEM-T Easy Vector using EcoRI and subcloned into a pCMV-Myc-N Vector (Clontech, California, USA). A corresponding pCMV-Myc-N vector was used as a negative control.

**Mutagenesis of FBP1 plasmids**. D119N, P120L, R158W G164S, A177D, G207R, N213K, G260R, E281K, P284R, G294E, G294V, and V325A mutations were introduced by site-directed mutagenesis (Quick Change Lightning Site-Directed Mutagenesis KIT, Agilent Technologies) using the primers listed in Supplementary Table 4. XL10-Gold ultracompetent cell DNA was isolated from cultured single clones and sequenced to confirm successful mutagenesis.

**Generation of FBP1 knockout HepG2 cells**. A hepatocellular carcinoma cell line (HepG2) was used in this study. HepG2 was purchased from the American Type Culture Collection. The cell line was tested for mycoplasma contamination. The protocol used for the CRISPR/Cas9 system was based on that reported by Cong et al.[25]. The backbone vectors pX459 pSpCas9(BB)-2A-Puro and pX462 pSpCas9n(BB)-2A-Puro were obtained from Addgene (Massachusetts, USA). The target guide RNA sequences were designed for exon 5, including c.491G, exon 6, including c.581T, and exon 8, including the Japanese common mutation site (c.960-961insG), of the *FBP1* genome (5′-GCAGCCGGCTACGCACTGTA-3′, 5′-GCACCAAAATGAACTCCCCGA-3′, and 5′-GTCGGGGGGATCCCAAGATCAC-3′)[3]. To clone the exon 5 and exon 6 target sequences into pX462 and the exon 8

target sequence into the pX459 backbone, oligos were synthesized using Eurofins genomics (Tokyo, Japan) (Supplementary Table 5). Then, these oligos were submitted to annealing and phosphorylation by means of a T4 DNA Ligase Reaction Buffer and a T4 Polynucleotide Kinase (New England Biolabs, Massachusetts, USA) used at 37 °C for 30 min and 95 °C for 5 min. pX459 and pX462 were digested using BbsI (Thermo Scientific, Massachusetts, USA) at 37 °C for 30 min, with gel purification being performed by use of a QIAquick Gel Extraction Kit (QIAGEN, Hilden, Germany). Ligation reactions of pX459, pX462, and the annealed oligos were performed for 10 min at room temperature using a Quick Ligation Kit (New England Biolabs, Massachusetts, USA). Then, the ligated oligos were purified using PlasmidSafe exonuclease (Cambio, Cambridge, UK) at 37 °C for 30 min. The plasmids were transfected into Stbl3, and the appropriate transfectants were amplified and collected using NucleoBond Xtra Midi (Takara, Kusatsu, Japan). Three plasmids (pX462-exon5 gRNA, pX462-exon6 gRNA, and pX459-exon8 gRNA) were cotransfected into HepG2 cells using Lipofectamine 3000 (Thermo Scientific, Massachusetts, USA) according to the manufacturer's protocol. The plasmid-expressing HepG2 cells were selected by puromycin (Wako, Osaka, Japan), and the limiting dilution method was used to establish the monoclonal cell line.

**Cell culture and transient transfection**. The human hepatocarcinoma cell line *FBP1*-KO HepG2 was cultured using Dulbecco's modified Eagle's medium containing antibiotics and 10% fetal bovine serum. *FBP1*-KO HepG2 cells were plated on 6-well plates and then transfected with plasmid DNA (2.5 μg) complexed with Lipofectamine 3000 reagent (7.5 μl) and P3000 reagent (5 μl) in 250 μl of Opti-MEM (Thermo Scientific, Massachusetts, USA). Subsequently, the FBPase activity of HepG2 cells was measured 48 hours after transfection using a nicotinamide adenine dinucleotide phosphate (NADP)-coupled spectrophotometric assay.

**FBPase activity assay**. The FBPase activity was calculated from an NADP-coupled spectrophotometric assay as described by Kikawa et al.[26]. The assay mixture (300 μl) was composed of 40 μg protein of cell lysate, 50 mmol/L Tris-HCl buffer (pH 7.5), 2.0 mmol/L MgCl₂, 1.0 mmol/L EDTA, 0.2 mmol/L NADP, 3.5 U/mL glucose-6-phosphate dehydrogenase, 1.5 U/mL glucose-6-phosphate isomerase, and 100 μmol/L FBP, which served as the substrate. A 96-well plate reader was used to record the rate of NADPH formation.

**Quantitative reverse transcription PCR (RT-qPCR) analysis**. RT-qPCR experiments were performed as previously described[27–29]. The *FBP1* gene-specific mRNA expression values were determined and normalized to those of β-actin as an internal control. Briefly, total RNA (4 μg) was extracted using an RNeasy kit (Qiagen, California, USA) and reverse-transcribed using a ReverTra Ace qPCR RT Kit (Toyobo, Tokyo, Japan). The cDNA products were subjected to RT-PCR using a Step One Plus Real-Time PCR system (Applied Biosystems, Massachusetts, USA). All primer information is provided in Supplementary Table 6.

**Immunoblot analysis**. Immunoblot analyses were performed as previously described[30]. Briefly, ECL-films were used for western blot developing. After the optimal exposure time, we open the cassette and pop the film directly into the film developer. Once our film has been developed, we overlay it back on the blot to mark the protein ladder on the film with a Sharpie marker. Western blot films with size marker labeled by a Sharpie marker as uncropped western blots are provided in Supplementary Fig. 6. In parallel with the ECL-films, another immunoblot was performed under the same conditions as Fig. 2a using a CCD imager Fusion FX (M&S Instruments Inc., Osaka, Japan) to obtain accurate band size information (Supplementary Fig. 2). We confirmed that the Myc, and actin protein bands and size markers using Fusion FX were consistent with those labeled by a Sharpie marker using ECL-films. The antibodies included those against FBP1 (SIGMA rabbit polyclonal, clone: HPA005857), c-Myc (Santa Cruz mouse monoclonal, clone: 9E10), actin (SIGMA rabbit polyclonal, clone: A2066), ATF6 (Novus Biologicals mouse monoclonal, clone: 70B1413.1), HSP70 (StressMarq mouse monoclonal, clone: N27F3-4), HSP90 (Santa Cruz mouse monoclonal, clone: sc-13119), HSP60 (Abcam rabbit polyclonal, clone: ab46798) and TCP1 (Bethyl Laboratories rabbit polyclonal, clone: A303-444A). The ratio of cells with FBP1 aggregates (%) was counted by independent researchers.

**Endoplasmic reticulum (ER) fractionation**. The extraction of the ER was performed using an Endoplasmic Reticulum Enrichment Kit (Novus Biologicals, Colorado, USA) according to the manufacturer's instructions.

**Immunofluorescence analysis**. Immunofluorescence analysis was performed to examine the cellular expression of WT and mutant FBP1. *FBP1*-KO HepG2 cells were cultured in 4-well chamber slides ($2 \times 10^4$ cells/well) and then transfected with plasmids (0.5 μg). Twenty-four hours after the transfection, the cells were treated with CellLight® ER-GFP (12 μl/well; Thermo Scientific, Massachusetts, USA), which is a marker of ER. This was followed by fixation in 100% ethanol at −20 °C for 10 min and incubation with blocking solution and primary antibodies against FBP1 (SIGMA rabbit polyclonal, clone: HPA005857), c-Myc (Santa Cruz mouse

monoclonal, clone: 9E10) and GFP (MBL rabbit polyclonal, clone: 598) 48 hours after the transfection. A confocal laser microscope (LSM710, Carl Zeiss, Germany) was used to obtain fluorescence images.

**Immunoprecipitation assay**. The commercially available c-Myc-tagged Protein Mild Purification Kit (Medical & Biological Laboratories, Aichi, Japan) was used to perform the immunoprecipitation assay. HepG2 cell extracts containing different Myc-tagged FBP1 variants were incubated with anti-Myc beads at 4 °C for 1 hour. The beads were then rinsed and eluted using a wash solution and elution peptides.

**Mass spectrometry sample preparation**. *FBP1*-KO HepG2 cells were transfected with plasmid DNA containing Myc-tagged FBP1-WT, Myc-tagged FBP1-G164D, or Myc-tagged FBP1-F194S. These cells were treated with the mannosidase inhibitor kifunensine (200 μM) for 48 h. Myc-tagged FBP1 HepG2 cell solubilization was obtained with the following buffer: 50 mM Tris-HCl (pH 7.5), 1.0 mM MgCl₂, 0.1 mM EDTA, 0.5 mM phenylmethanesulfonyl fluoride, 2.5 μg/mL leupeptin and 1.0 μg/mL antipain. Immunoprecipitation with anti-Myc beads (Medical & Biological Laboratories, Aichi, Japan) was performed at 4 °C for 1 h. The beads were then rinsed and eluted using a wash solution and elution peptides. Immune complexes were separated by SDS-PAGE. Bands were removed from the gel and examined by mass spectrometry to find corresponding proteins. Respective gel pieces were rinsed two times using 100 mM bicarbonate in acetonitrile with subsequent protein digestion by trypsin. Then, 0.1% formic acid was added to the supernatant, and the peptides were subjected to liquid chromatography-tandem mass spectrometry (LC-MS/MS) using an LTQ Mass Spectrometer (Thermo Scientific, Massachusetts, USA). Analysis of the MS/MS dataset results was conducted using the Mascot software program (Matrix Science, Massachusetts, USA). The source Mass spectrometry data is shown in the Supplementary Data 2.

**Proteomic analysis and database search**. Proteomic analysis and database searches were performed as previously described[31]. Kyoto Encyclopedia of Genes and Genomes (KEGG) pathway annotation and Gene Ontology (GO) analyses were performed using the STRING interaction database.

**Protein structure based on the protein data bank**. Ke et al. revealed the detailed crystal structure of *Sus scrofa* FBP1 complexed with fructose 6-phosphate, AMP, and magnesium[32]; Fig. 1g shows the structure of the FBP1 dimer based on their report (DOI: 10.2210/pdb1FBP/pdb). The FBP1 amino acid sequences of *Homo sapiens* and *Sus scrofa* show approximately 90% similarity. In particular, the functional motifs directly associated with FBPase activity are highly conserved.

**Statistics and reproducibility**. The results are presented as the mean ± SD. One-way ANOVA followed by Dunnett's multiple comparison test was used to analyze continuous variables. Significance is indicated by asterisks or NS, as follows: *$P < 0.05$, **$P < 0.01$; NS: not significant $P > 0.05$.

The experimental findings were reliably reproduced with at least two independent experiments in Figs. 1f, 2a, b, g–k, 3b–d, 4a–c.

**Study approval**. The genetic study and collection of patient information were approved by the ethics committee of Chiba University Graduate School of Medicine. Written informed consent for the genetic studies was obtained from the patient and her family in accordance with the protocols issued by the Chiba University Graduate School of Medicine. All relevant ethical regulations were followed for experiments with human participants.

**Reporting summary**. Further information on research design is available in the Nature Portfolio Reporting Summary linked to this article.

## Data availability

Data supporting the findings of this study are available within the article and its Supplementary data files. Source data for all the figures are provided in a single Excel file as a Supplementary Data 3. Uncropped and unedited bot images are available as Supplementary Fig. 6. The whole-exome sequencing data is available in the National Bioscience Database Center (NBDC) Human Database (http://humandbs.biosciencedbc.jp/en) with the accession number of JGAS000625. The Japanese Genotype-phenotype Archive (JGA) services are provided in collaboration with the Department of NBDC Program (NBDC) of Japan Science and Technology Agency. Briefly, the users can apply for data access through the NBDC application system after logging in with their D-way account. In the application, create a data user group, specify JGA Study and Dataset accessions, and register a public key for dataset decryption. After application is approved, access to the JGA server with D-way account and donwload data to on-/off-premise servers by WinSCP or sftp. Encrypted data files and decryption tools are provided, decrypt the data files by using the private key paired with the public key for dataset decryption registered in the application. The mass spectrometry proteomics data was deposited in the ProteomeXchange Consortium (http://proteomecentral.

proteomexchange.org) via the jPOST partner repository (http://jpostdb.org) with the dataset identifier PXD042364. Source data are provided with this paper.

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

## Acknowledgements
The authors thank Yue Yao and Sae Yano for their excellent technical assistance. This work was supported by grants from the Ministry of Education, Culture, Sports, Science and Technology (Japan) [Grants-in-Aid: for Scientific Research (B) #21H02974, #19H03708, #22300325; (C) #22K08644, #22K07205, #22K08619 #21K07145, #21K08524, #20K08397, #20K07561, #19K07635, 19K08972, #18K07439, #18K08464; Challenging Research (Exploratory) #21K19398; Early-Career Scientists #20K17527, #19K17999, #17K16160; Fund for the Promotion of Joint International Research (Fostering Joint International Research (A); #19KK04071, #20KK0373, #22KK0271; T.T. was supported by Japan Society for the Promotion of Science KAKENHI grant JP19H03708. This work was partly supported by The Uehara Memorial Foundation, Mochida Memorial Foundation for Medical and Pharmaceutical Research, The Naito Foundation, Mitsui Life Social Welfare Foundation, Princes Takamatsu Cancer Research Fund, Takeda Science Foundation, Senshin Medical Research Foundation, Japan Diabetes Foundation, Yamaguchi Endocrine Research Foundation, The Cell Science Research Foundation, The Ichiro Kanehara Foundation for the Promotion of Medical Sciences and Medical Care, the Yasuda Memorial Medical Foundation, MSD Life Science Foundation, The Hamaguchi Foundation for the Advancement of Biochemistry, The Novartis Foundation (Japan) for Promotion of Science, Kose Cosmetology Research Foundation and the Medical Institute of Bioregulation Kyushu University Cooperative Research Project Program.

## Author contributions
I.S., H.N., T.Mi., and T.T. designed the study. I.S., H.N., M.F., Y.T., T.Ma., H.A., S.K., T.K., M.N., and K.Y. collected clinical information and generated mutant vectors, and conducted experiments. Y.F., T.O., and S.O. performed genomic analysis. M.F., A.N., and T.T. performed proteomics analysis. I.S., H.N., M.F., and M.Y. performed bioinformatics and statistical analyses. I.S., N.H., M.F., T.F., and A.N. performed immunohistochemistry and histopathologic analyses. I.S., H.N., N.H., M.F., E.L., T.Mi., and T.T. analyzed, discussed, and interpreted the data. I.S., and T.T. wrote the manuscript. T.T. coordinated and directed the project. All authors approved the submitted manuscript.

## Competing interests
The authors declare no competing interests.
