## [Peer Review File · Communications Biology]

Reviewers' comments:

Reviewer #1 (Remarks to the Author):

The scientific content of the manuscript is great. The methodology and results were well described and elaborated. Classifying mutations into 3 categories was interesting. However some part of the text requires professional English language editing (highlighted in yellow). The conclusion is too lengthy.

Reviewer #2 (Remarks to the Author):

The authors present a case of FBPase deficiency with genotype-phenotype correlation. They have further performed interactome analysis and attempted localization of FBPase protein and an attempt to classify previously reported missense variants. The experiments are comprehensive, well-reported and the conclusions appear logical.

Major points:

1. The G294V variant has been grouped as variant without change in hydrophobicity, even though they have acknowledged significant change in hydrophobicity in wild vs mutant amino acid in table 6. They may reconsider classifying the variant into the group with change in hydrophobicity, where most of the variant properties are better correlated.

Minor points:

Conclusion needs a revision and should be focussed on the results of present study only. Statement in line 469 ' Thus, G207V and V325A were defined as Type 3 mutations' appears more logical when placed after line 467 ' Kikawa et al. suggested that this mutation does not play a pathogenic role in FBPase deficiency, which is consistent with findings of our study'

Reviewer #3 (Remarks to the Author):

The authors focused on genotype - biochemical phenotype characterisation of fructose-1,6-4 bisphosphatase deficiency performing sequencing approaches with function studies. Early diagnosis is very important in FBPase deficiency therefore, authors should cite the article <https://doi.org/10.1515/tjb-2019-0473>.

Besides that after this minor revision, this contributing study should be accepted.

Response to reviewers

Reviewer #1:

Comments:

The scientific content of the manuscript is great. The methodology and results were well described and elaborated. Classifying mutations into 3 categories was interesting. However some part of the text requires professional English language editing (highlighted in yellow). The conclusion is too lengthy.

Reply:

Thank you for your thoughtful comment. We concisely revised the conclusion part's description (line 512-534) and checked it with professional English language editing.

Reviewer #2:

Comments:

The authors present a case of FBPase deficiency with genotype-phenotype correlation. They have further performed interactome analysis and attempted localization of FBPase protein and an attempt to classify previously reported missense variants. The experiments are comprehensive, well-reported and the conclusions appear logical.

Major points:

1. The G294V variant has been grouped as variant without change in hydrophobicity, even though they have acknowledged significant change in hydropathy in wild vs mutant amino acid in table 6. They may reconsider classifying the variant into the group with change in hydrophobicity, where most of the variant property are better correlated.

Reply:

Thank you for this excellent suggestion. Both glycine and valine are classified into hydrophobic amino acids. Therefore, the G294V variant was grouped as a variant without change in hydrophobicity in figure 3. As the reviewer suggested, there is a certain change in the hydropathy index between glycine (-0.4) and valine (4.2) in Table 1, even though glycine and valine are hydrophobic amino acids. This hydropathy difference may affect FBP1 protein misfolding. We described the hydropathy difference as the possible explanation for G294V showing type 2 features (line 494-498). Thank you for your important comment.

Minor points :

Conclusion needs a revision and should be focussed on the results of present study only. Statement in line 469 ' Thus, G207V and V325A were defined as Type 3 mutations' appears more logical when placed after line 467 ' Kikawa et al .suggested that this mutation does not play a pathogenic role in FBPase deficiency, which is consistent with findings of our study'

Reply:

Thank you for your thoughtful comment. We revised the description as the reviewer suggested (line 500-508). This description was moved to the discussion part to shorten the conclusion part.

Reviewer #3 :

The authors focused on genotype - biochemical phenotype characterisation of fructose-1,6-4 bisphosphatase deficiency performing sequencing approaches with function studies. Early

diagnosis is very important in FBPase deficiency therefore, authors should cite the article <https://doi.org/10.1515/tjb-2019-0473>.

Reply:

Thank you for your important suggestion. We cited the article in the manuscript (line 445-446).

REVIEWERS' COMMENTS:

Reviewer #2 (Remarks to the Author):

The reviewed version appears to be well-written and has addressed the questions raised by the reviewer satisfactorily. The article presents a comprehensive functional study of FBP1 variants in a scientific manner.

The article is recommended to be accepted for publication.

Response to reviewers

The Reviewer #2 (Remarks to the Author):

The reviewed version appears to be well-written and has addressed the questions raised by the reviewer satisfactorily. The article presents a comprehensive functional study of FBP1 variants in a scientific manner.

The article is recommended to be accepted for publication.

Reply:

Thank you for your reviewing our study, and thoughtful comments.

We are very happy to hear that our paper is now recommended to be accepted for publication.